# Initial Treatment with Biological Therapy in Rheumatoid Arthritis

**DOI:** 10.3390/jcm13010048

**Published:** 2023-12-21

**Authors:** Jesús Tornero Molina, Blanca Hernández-Cruz, Héctor Corominas

**Affiliations:** 1Departamento de Reumatología, Hospital de Guadalajara, 19002 Guadalajara, Spain; 2Departamento de Medicina y Especialidades Médicas, Universidad de Alcalá, 28805 Madrid, Spain; 3Departamento de Reumatología, Hospital Universitario Virgen Macarena, 41009 Sevilla, Spain; blancahcruz@gmail.com; 4Departamento de Reumatología, Hospital Universitari de Sant Pau & Hospital Dos de Maig, 08025 Barcelona, Spain; vancor@yahoo.com; 5Medicine Faculty, Universitat Autònoma de Barcelona (UAB), 08193 Barcelona, Spain

**Keywords:** rheumatoid arthritis, biologic therapy, initial treatment, narrative review, experts’ opinion

## Abstract

Background: We aimed to analyse the effectiveness, efficiency, and safety of initial treatment with biological therapies in rheumatoid arthritis (RA). Methods: Qualitative study. A group of RA experts was selected. A scoping review in Medline was conducted to analyse the evidence of initial RA treatment with biological therapies. Randomised clinical trials were selected. Two reviewers analysed the articles and compiled the data, whose quality was assessed using the Jadad scale. The experts discussed the review’s findings and generated a series of general principles: Results: Seventeen studies were included. Most of the included patients were middle-aged women with early RA (1–7 months) and multiple poor prognostic factors. Initial treatment with TNF-alpha inhibitors combined with methotrexate (MTX) and an IL6R inhibitor (either in mono or combination therapy) is effective (activity, function, radiographic damage, quality of life), safe, and superior to MTX monotherapy in the short and medium term. In the long term, patients who received initial treatment with biologicals presented better results than those whose initial therapy was with MTX. Conclusions: Initial treatment of RA with biological therapies is effective, efficient, and safe in the short, medium, and long term, particularly for patients with poor prognostic factors.

## 1. Introduction

Rheumatoid arthritis (RA) is a chronic inflammatory joint disease of autoimmune origin, with an estimated prevalence of around 1% [1]. The disease is associated with a major impact on the patient, their environment, and the healthcare system [2,3,4].

For many years, the pharmacological treatment of RA was essentially based on the use of corticosteroids and classical synthetic disease-modifying anti-rheumatic drugs (csDMARD) like methotrexate (MTX) [5]. However, the arrival of biologic DMARD (bDMARD) significantly changed treatment paradigms for RA patients [6].

Pivotal studies on bDMARD showed their efficacy and safety for patients with RA refractory to csDMARD [5,7], which led regulatory agencies to approve the use of these drugs for this group of RA patients. Subsequent publications have presented the results of *post hoc* analyses and studies specifically designed to analyse bDMARD as the initial treatment for RA (csDMARD-naïve patients) [8,9]. This is why the summaries of products for some bDMARD contemplate the drug’s use in patients with severe, active, progressive RA not previously treated with MTX or other csDMARD. However, the guidelines of national and international scientific societies still recommend using csDMARD, particularly MTX, as the initial treatment for RA [6,10]. These guidelines highlight the high cost of many bDMARDs as a limitation to their wider use [11,12].

On the other hand, the approval of biosimilar medicines has ushered in more competitive pricing, bringing a considerable reduction in bDMARD costs [13]. A report by BioSim (the Spanish Association of Biosimilar Medicines) estimates savings derived from the introduction of biosimilars for the 2009–2019 period to be 2306 million euros in Spain [14]. It also highlights adalimumab (ADA) as the drug that generates the most savings [14]. There are currently over 60 biosimilar medicines corresponding to 17 drugs, 4 of which are for immune-mediated diseases: infliximab (IFX), ADA, etanercept (ETN), tocilizumab (recently approved by the European Medicines Agency), and rituximab, which are widely used [15]. The guidelines of scientific societies acknowledge that biosimilar medicines have contributed to a substantial reduction in the cost of medicines, following a policy of rational prescribing based on the principle that if two medicines are equally effective and safe for a specific patient, the least expensive should be used [6].

In light of this, we can consider the use of bDMARD as the initial treatment for RA, at least for some subgroups of patients. The design of this project aimed to address this topic, with the objectives of analysing the existing evidence on initial treatment with bDMARD and issuing a series of positions for better stratification of RA patients. Bearing in mind that many countries recommend biosimilar TNF-alpha inhibitors as the first choice for bDMARD and, among patients with intolerance to/contraindications for MTX, an interleukin 6 receptor (IL6R) inhibitor, this document focuses on these pharmacological groups. Abatacept and rituximab are currently reserved for special situations, and in clinical practice, Janus kinase inhibitors are used in later lines of treatment.

We believe that this document will help rheumatologists in their therapeutic decision-making for RA patients.

## 2. Materials and Methods

This qualitative study is based on a scoping review of the literature and on expert opinion. The project was conducted in full compliance with the principles established in the Declaration of Helsinki for medical research involving human subjects in its latest version [16] and in accordance with applicable regulations on Good Clinical Practice.

Participant selection and the first nominal group meeting. A group of three rheumatologists with extensive experience and knowledge of RA management were selected. In a first meeting, they analysed the current status of RA management and defined the objectives, scope, and literature review to be undertaken.

Scoping review of the literature. The objective was to assess the efficacy, effectiveness, and safety of the initial treatment of RA with bDMARD. 

With the help of an expert documentalist, different search strategies were designed for Medline (up to January 2023) that combined both MeSH and free-text terms. Examples of terms used are “rheumatoid arthritis” or “disease-modifying anti-rheumatic drugs”. Searches were also made using PubMed’s Clinical Queries tool (see Appendix A).

Studies that met the following inclusion and exclusion criteria were selected: The population included patients with RA (according to international criteria [17]), adults (>18 years), regardless of disease duration or severity (P); RA patients receiving initial treatment with bDMARD, i.e., naïve for any csDMARD such as oral (po), subcutaneous (sc), or other MTX. No restrictions were imposed regarding dose, treatment duration, use as monotherapy or in combination (I), a control, or either a placebo or with an active ingredient (C). We also included studies that analysed any variable related with efficacy/effectiveness, such as RA activity including DAS28 (disease activity score), CDAI (clinical disease activity index), SDAI (simplified disease activity index), disease remission, function assessed with the HAQ (Health Assessment Questionnaire), structural damage viewed in a simple radiography or magnetic resonance imaging (MRI), PRO (patient-reported outcomes), acute phase reactants such as erythrocyte sedimentation rate (ESR) or C-reactive protein (CRP), quality of life assessed with RAQoL (Rheumatoid Arthritis Quality of Life) Questionnaire or others, cost-effectiveness and safety variables (serious adverse events, infection rate, etc.). The types of studies accepted were randomised clinical trials (RCT) with their corresponding extension studies and post hoc analyses. Figure 1 shows the study flow chart.

All of the citations found using the different search strategies were downloaded to the EndNote^®^ reference management software package (Version number 20). Two reviewers (EL, TO) independently analysed the citations in duplicate. The search results were first refined by title and abstract, with another purge after a detailed reading of the resulting citations. The two reviewers compiled data from the included studies. To assess the methodological quality of the studies included, a Jadad scale was used (from 1 to 5, with “quality” defined as an RCT with a score of ≥3). A descriptive-qualitative analysis was performed. Meta-analysis was only performed in cases of homogeneity.

Finally, as part of the secondary search, the bibliographic references of the included articles and the abstracts of international conferences were reviewed.

Second meeting of the nominal group. The nominal group discussed the review results at the second meeting, during which the experts reached a consensus on a series of points related to the use of bDMARD as an initial treatment for RA.

Preparation of the final document. The final text took into account both the narrative review and the nominal group’s decisions. For each of the expert points, we assessed the strength of the recommendation (very low, low, moderate, or high). The resulting document was given to the experts for their final evaluation and comments.

## 3. Results

A total of 17 articles [18,19,20,21,22,23,24,25,26,27,28,29,30,31,32,33,34] were included, whereas 15 were excluded (see Appendix A) [8,9,35,36,37,38,39,40,41,42,43,44,45,46,47]. A summary of the main outcomes is depicted in Table 1, while the Appendix A provides the evidence tables (Appendix A).

Here we present the expert opinions along with the literature search results.

General indications on initial treatment with bDMARD.

Based on the body of evidence and the current context of daily practice, it is possible to consider the use of bDMARD as an initial treatment for RA. Strengths of the recommendation: Moderate

With regard to TNF-alpha inhibitors, one RCT found that the combination IFX + MTX po was significantly superior to MTX po in improving synovitis and bone marrow oedema (measured by MRI), both at weeks 18 and 52 [19]. The combination was also superior in ACR20/50/70 responses at week 22, with a similar (non-significant) trend observed at week 52 [19]. 

Data from the *BeSt* study show that the triple therapy of csDMARD with prednisone and the combination IFX + MTX (po and sc) improved HAQ faster (3rd month) than the other treatment groups analysed, an improvement that was sustained until month 12 (*p* < 0.05). Moreover, these strategies were also significantly superior in improving structural damage (<0.001), although this effect disappeared in the second year [20,21,22]. Long-term (10-year) follow-up showed that the different combination therapies produced similar global outcomes (non-significant differences), although patients treated with IFX + MTX showed less radiographic progression and better physical functioning than the other strategies [23]. 

In the IDEA RCT, the combination of initial therapy with IFX + MTX po or MTX + intravenous methylprednisolone achieved similar DAS28 remission rates at week 78: 50% vs. 48% (*p* = 0.795); however, DAS28 remission was reached more quickly with IFX + MTX. Although both groups obtained a high proportion of patients without radiographic progression, no differences were detected [24]. 

The HIT-HARD [26], GUEPAR [27], and OPERA [28] RCTs evaluated ADA as the initial treatment for RA. The outcomes of the HIT HARD study at week 24 showed that ADA + MTX sc was significantly superior to MTX sc in terms of DAS28, HAQ, and DAS28 remission (47.9% vs. 29.5%; *p* = 0.021), as well as in ACR 50 and ACR 70 responses [26]. From week 24, all patients received MTX sc monotherapy. Radiographic progression at week 48 was significantly greater in patients who had received MTX sc monotherapy as the initial treatment [26]. The OPERA study compared ADA po + MTX vs. MTX po [28]. At one year of treatment, the combination ADA + MTX was significantly superior to MTX in terms of DAS28-CRP, CDAI, SDAI, HAQ, ACR/EULAR28 and ACR/EULAR40, ACR50 and ACR70 responses, and in remission, defined with DAS28, CDAI, SDAI, and ACR/EULAR. The SF-12 questionnaire revealed a significant improvement in quality of life for physical but not mental health, which was echoed in the EQ-5D (*p* = 0.015) [28]. Finally, the GUEPAR study reported that initial treatment with ADA + MTX po was significantly superior at 3 months to MTX po in improving rigidity, ACR 20/50/70 response, good EULAR response (63% vs. 25%), and DAS28 remission (36% vs. 12%). No differences existed between the groups for pain, fatigue, or ESR [27].

For ETN, the EMPIRE study [29] included 110 patients with early-onset synovitis (41% met RA criteria according to 1987 ACR criteria and 94% according to 2010 ACR-EULAR criteria). At week 52 of treatment, 32.5% of patients with ETN + MTX po did not present painful or swollen joints vs. 28.1% with MTX po (no statistically significant differences). Nor were any differences found in ACR remission, SDAI remission, DAS44-CRP, HAQ-DI, SF-36, EQ5D-3L, radiographic progression, or in the variables assessed at week 72 [29]. 

There are 3 published good-quality RCTs on the initial treatment of RA with certolizumab pegol (CZP) [30,31,32]. The outcomes of *C-EARLY* show that the combination CZP + MTX po is significantly superior to MTX po in controlling disease activity, sustaining clinical response, functional improvement, and inhibiting radiographic progression. In week 52, 28.9% of patients in the CZP + MTX group reached DAS28 remission: 28.9% vs. 15% for MTX (*p* < 0.001) [30]. The study also revealed significant differences in favour of CZP + MTX in sustained low disease activity (DAS28 ≤ 3.2), ACR 50 response, functioning, inhibition of radiographic progression, and both CDAI and SDAI remission. The study went on to assess whether, for patients who had achieved sustained low disease activity after 1 year of treatment with standard doses of CZP (200 mg/2 weeks) + MTX po, continuing with a standard or optimised dose of CZP (200 mg/4 weeks) was superior to interrupting CZP (and continuing with MTX po) for an additional period of 1 year (a total of 104 weeks). At week 104, the proportion of patients with sustained low disease activity was higher in the CZP-treated group, both at standard and optimised doses (48.8% and 53.2%) than in patients who continued with MTX (39.2%), although the differences were only significant for the comparison of optimised dose vs. MTX (*p* = 0.041). The trend was similar for inhibition of radiographic progression and physical functioning [32]. 

There are 3 articles available for TCZ, based on 2 studies. One good quality study, *U*-Act-Early [33,34], and another moderate quality study [31]. TCZ as monotherapy or in combination with MTX po were superior to MTX po in different outcome measures. At week 24, almost double the number of patients receiving TCZ achieved DAS28 and CDAI remission compared with MTX (86% and 83% with TCZ + MTX and TCZ, respectively, vs. 44% with MTX, *p* < 0.0001). The therapies with TCZ were also significantly superior to MTX in terms of good EULAR response (89% vs. 87% vs. 49%), ACR (20/50/70/90 responses), and HAQ. In week 104, little progression was observed in all groups, but it was significantly lower in both TCZ arms than in the MTX po arm [34]. Patients in this study underwent an additional follow-up of 3 years in real life. The data for effectiveness were sustained, with no differences between the groups. During 5 years of follow-up, the accumulated time of sustained remission was significantly higher in the groups that began therapy with TCZ (median 216 and 190 days for the TCZ + MTX combination and TCZ, respectively) than for MTX (median 172). However, no differences were observed in radiographic progression between the groups [33]. 

2.Initial treatment with bDMARD should be considered, particularly for RA patients with a high inflammatory load and/or other poor prognostic factors. Strengths of the recommendation: High

Although RA is a heterogeneous, dynamic disease, different studies have shown that the baseline presence of certain factors is associated with a worse prognosis in terms of activity (e.g., remission) and radiographic damage [48]. Among these factors, we highlight a positive rheumatoid factor (RF) and/or anti–citrullinated protein antibodies, high counts of acute phase reactants, high disease activity (multiple inflamed joints, high activity index, etc.), significant functional limitation, or the presence of erosion in imaging tests [48]. 

Most patients included in the RCT analysed presented poor prognostic factors at baseline (see Table 2). The mean age of patients included was between 45 and 55 years, RF positivity was occasionally even higher than 90%, the mean DAS28 in most cases was higher than 5, and many patients presented structural damage in radiographic imaging [18,19,20,21,22,23,24,25,26,27,28,29,30,31,32,33,34]. 

On the other hand, early, aggressive treatment has been shown to improve the long-term prognosis of this disease [49]. This therapy is associated with higher clinical response rates, lower disability, and less structural damage [50,51]. Although a larger body of specific evidence is necessary, in the BeSt study’s 10-year follow-up, patients who received IFX + MTX showed less radiographic progression and better physical functioning than patients following other strategies [23]. The data at 5 years for TCZ are similar [33], as are the 10-year extension data for ADA of the PREMIER study (almost 30% of whose patients had previously received csDMARD) [52]. However, it is important to consider each case individually. The decision to use bDMARD as initial treatment should be discussed with the patient, evaluating the potential risks and benefits of these drugs on a case-by-case basis.

3.If considering a TNF-alpha inhibitor as an initial treatment for RA, its use is recommended in combination with MTX, whereas if considering an IL6R inhibitor, monotherapy is possible. Strength of the recommendation: Moderate

Most of the RCTs analysed included patients with MTX po either combined with a TNF-alpha inhibitor or as monotherapy [18,19,20,21,22,23,24,25,27,28,29,30,31,32]. The studies used different doses and regimens of MTX po, but the majority began with a dose of 7.5–10 mg/week, progressively increasing each 2–4 weeks to a maximum of 30 mg/week; however, most studies used a maximum dose of MTX between 15 and 20 mg/week.

As we have described, the use of MTX in combination with a TNF-alpha inhibitor is an effective and safe initial treatment for RA [18,19,20,21,22,23,24,25,27,28,29,30,31,32]. Although we have not found comparative evidence between TNF-alpha inhibitors for initial treatment as monotherapy and in combination with MTX, based on publications referring to patients with refractory RA, experts recommend the use of TNF-alpha inhibitors with MTX as the initial therapy [7]. In this context, as well as being more effective, MTX can prevent immunogenicity or reduce the use of corticosteroids [53]. On the other hand, MTX po might be recommended because it is the drug for which most evidence is available due to its lower price (cost-effectiveness). However, it is important to remember that the bioavailability of parenteral MTX is higher than that of MTX po, particularly at doses of ≥15 mg/week [54,55], and that in patients with an inadequate response to orally administered MTX (15 mg/week), dose scaling using parenteral MTX is more clinically effective [55,56]. Likewise, we should always consider patient opinion and preference when making therapeutic decisions. So, in some cases, we may consider the use of parenteral MTX with TNF-alpha inhibitors as the initial treatment.

On the other hand, there are considerable variations in the use of MTX in clinical practice, including the initial dose, dose scaling, maximum dose, etc. [57,58]. A sub-analysis of the AR Excellence study [59] highlighted these variations, finding that MTX scaling to its full dose did not occur as quickly as it should and that there was not correct use of parenteral MTX administration. 

The outcomes reported for the use of IL6R as initial treatment [33,34] are similar to those reported for patients with RA refractory to csDMARD, so its use can be considered a monotherapy. 

4.Initial treatment of RA with bDMARD in monotherapy or in combination therapy is safe. Strength of the recommendation: Moderate

Data on the safety of bDMARD as an initial treatment are similar to those reported in studies on RA refractory to csDMARD [18,19,20,21,22,23,24,25,26,27,28,29,30,31,32,33,34]. It should be noted that the data analysed is for RCTs with different follow-up periods and even changes in treatment within the same study.

Infections are still the most frequent adverse event with the use of bDMARD as the initial treatment for RA, although the percentage of severe infections is generally low and similar to existing reports [18,19,20,21,22,23,24,25,26,27,28,29,30,31,32,33,34]. No new signs related to the active infection by hepatitis or tuberculosis viruses were detected. Haematological adverse events are not very frequent, and when they do occur, they are mild and reversible. An increase in liver enzymes is a relatively frequent adverse event, but in most cases, it is mild [18,19,20,21,22,23,24,25,26,27,28,29,30,31,32,33,34]. There are exceptional reports of severe cases but no clearly established causal relationship. Based on the data from the included studies, the use of bDMARD as an initial treatment cannot be associated with the development of any type of cancer [18,19,20,21,22,23,24,25,26,27,28,29,30,31,32,33,34]. A larger number of specific, long-term studies on patients in real life are necessary to analyse this issue in greater depth. 

On the other hand, the global percentage of adverse events of combinations of bDMARD with MTX in some studies is superior to that of MTX monotherapy, specifically for certain types of adverse events such as infections [33], but combination has not been associated with a higher risk of serious adverse events of any type. 

5.Initial treatment of RA with biosimilars of TNF-alpha inhibitor biological drugs is cost-effective and this specific variable should be included in treatment stratification. Strength of the recommendation: Moderate

Clinical variables, along with patient opinion and preferences, are essential elements in RA therapy decision-making. However, we must not forget the current healthcare context and its sustainability. This is why the experts consider that the cost-effectiveness of bDMARD is another important variable to take into account. 

Different reports at international conferences have informed us about studies into the cost-effectiveness of biological therapy treatment in Spain [60,61,62]. As no effectiveness difference was seen, a cost minimization analysis was performed that showed that currently the most cost-effective option for immune-mediated diseases such as Crohn’s or RA is ADA [60,61]. Specifically in the case of RA, it represents the cheapest biological therapy, with an estimated annual cost of 4650 euros vs. the 4650–10,000 euros for other treatments [60].

In line with these data, a study on cost-effectiveness in Norway [63], where the healthcare system promotes the use of biosimilars to reduce costs, found that since the introduction of biosimilars, the number of RA patients receiving treatment with bDMARD and small molecule inhibitors has increased from 39% in 2010 to 45% in 2019. The proportion of patients who reached DAS28 also increased from 42% to 67%. However, the annual cost of treating a patient with these therapies decreased by 47% [63]. 

Another pharmacoeconomic study from the perspective of the Italian national health system found that for psoriasis patients, there is a high similarity in cost per responder with two ADA biosimilars (MSB1102 and ABP 501) compared with MTX sc, with cost-effectiveness closest at 52 weeks between MSB1102 (799 euros) and MTX sc (625 euros) [64]. 

Finally, a cost-effectiveness study on the efficiency of early treatment with biologicals [65] found that for RA patients refractory to csDMARD, adding a biosimilar TNF-alpha inhibitor to MTX at 6 months increased the total treatment cost by only £70, compared to continuing MTX monotherapy and waiting until 12 months (price in pounds sterling for 2017) [65].

## 4. Discussion

This project shows that the initial RA treatments of TNF-alpha inhibitors combined with MTX and of IL6R (as monotherapy or in combination) are effective, efficient, and safe, with data for effectiveness that in some cases surpasses csDMARD results [18,19,20,21,22,23,24,25,26,27,28,29,30,31,32,33,34]. 

For several reasons, including effectiveness, safety, cost-effectiveness, type of healthcare system, etc., the current strategy consists of starting treatment with csDMARD, habitually as a monotherapy. This review highlights that, in many cases, patients who began an initial treatment with bDMARD presented better outcomes (considering activity, physical functioning, PROs, quality of life, and image) than those who had done so with csDMARD [18,19,20,21,22,23,24,25,26,27,28,29,30,31,32,33,34]. This suggests that the option should be considered in clinical practice. 

It should be noted that most of the studies we excluded had a significantly high percentage (between 20 and 30%) of patients included who had previously used csDMARD [37,38,39,40]. This population may have characteristics and/or responses that are different from those of csDMARD-naïve patients. The patients included in this review were experiencing early-stage RA and had more than one variable for poor prognosis, and it is particularly for this subset of patients that initial treatment with bDMARD should be considered [18,19,20,21,22,23,24,25,26,27,28,29,30,31,32,33,34] or at the very least they should follow the current T2T strategy, changing stage in a short period of time (a maximum of 3 months) to more effective therapeutic options. To the best of our knowledge, there have been no similar initiatives in this field, and we consider that our results may be of interest in clinical practise to manage the initial treatment of RA patients.

It is striking that national and international consensus documents do not consider the use of bDMARD as an initial treatment for RA, citing various reasons, the foremost of which is cost [10]. Bearing in mind the current context, in which the arrival of biosimilars has brought a considerable reduction in the net cost of bDMARD [13], this position should be reconsidered. But taking into account the risks with the use of bDMARDs, we consider it very important to individualise the use of bDMARD as an initial treatment, trying to select patients that might benefit most. In our opinion, this treatment strategy might be particularly effective for RA patients with poor prognostic factors. 

The evidence for initial treatment with ETN deserves particular comment, although we can only draw conclusions based on one study, EMPIRE [29], which limits the robustness of our conclusions. The study did not find differences between the combination ETN + MTX and MTX monotherapy that were as consistent as those found in studies of other biologicals. There are different factors that could explain these results. The study included 110 patients, which is probably a small sample size (the calculation of the sample size was based on radiographic criteria, not on RA activity, as in other RCTs). The patients presented early-onset synovitis (some did not meet RA criteria), with data on activity and other severity factors that were lower than the other studies included. This RCT also allowed the use of MTX at comparatively higher doses than in other RCTs. All of these factors could cause the study not to have reached statistical significance in many of the outcome variables.

Another point to note is that recommendations include the use of MTX po (vs. MTX sc) in combination with bDMARD, both because more evidence is available and due to cost. However, the option of MTX sc should be evaluated case by case since its bioavailability is greater and the patient may prefer this route of administration [54,55].

Among the limitations of our work, we note that some of the studies included are post-hoc analyses or exploratory studies, so their outcomes should be interpreted with caution. Likewise, the inclusion of RCT means that the population consists of patients who are not 100% representative of the general RA population seen in conventional clinical practice. Finally, the lack of robust comparative data between combination therapy and monotherapy of biologicals could weaken our findings.

## 5. Conclusions and Future Directions

Initial treatment with bDMARD in RA patients is effective, has an acceptable safety profile, and, in our current context, should be considered, particularly for patients with poor prognostic factors. The final decision on RA treatment should be made on a case-by-case basis and in consensus with patients.

On the other hand, more research is needed in order to expedite and optimise treatment stratification by using advanced integrative modelling of complex health data (genetics, biomarkers, clinical data, etc.).

## Figures and Tables

**Figure 1 jcm-13-00048-f001:**
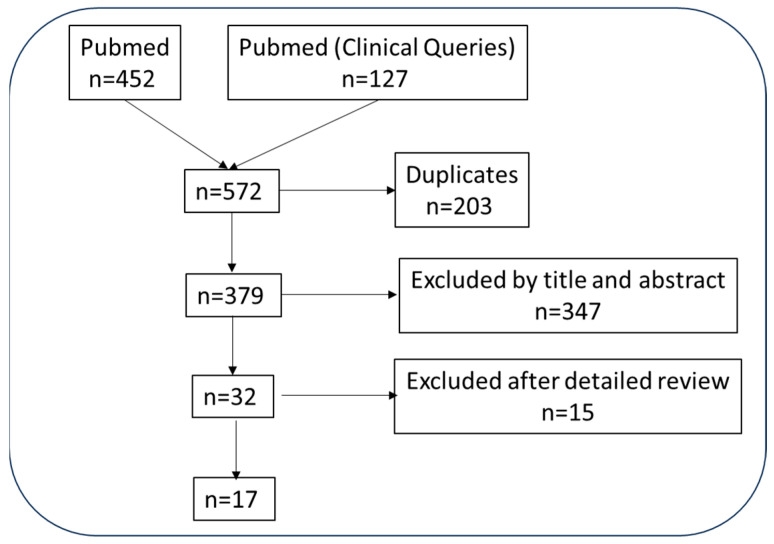
Studies flow chart.

**Table 1 jcm-13-00048-t001:** Main conclusions of the included studies [18,19,20,21,22,23,24,25,26,27,28,29,30,31,32,33,34]. The study’s primary outcomes are highlighted.

1	The use of initial treatment with IFX + MTX is significantly superior to MTX (in the short and long term) to MTX regarding RA activity (DAS28, DAS28 remission, imaging techniques, etc.), HAQ, radiographic progression, and quality of life (RAQoL)
2	Initial treatment with IFX + MTX is superior to other strategies: sequential csDMARD monotherapy and stepped combination treatment with csDMARDs regarding the previously mentioned variables
3	Initial use of ADA + MTX during the first year is significantly superior (in the short and long term) to MTX regarding RA activity (DAS28, DAS28/CDAI/SDAI/ACR-EULAR remission, low disease activity, joint counts, etc.). HAQ, quality of life (SF-36-physical component), and probably regarding radiographic progression
4	Current evidence on initial treatment with ETN + MTX is based on a single study and is inconclusive
5	Initial use of CZP + MTX is significantly superior (in the short and long term) to MTX regarding RA activity (DAS28, DAS28 remission, CDAI, CDAI remission, SDAI, acute phase reactants, etc.), HAQ, and radiographic progression. However, the CZP + MTX combination has not been shown to be superior to the combination of MTX with prednisone or to the combination of SSZ + HCQ + intra-articular corticosteroids
6	Initial use of ABT + MTX achieves remission rates of 52% at 24 weeks, higher than those obtained with the combination of MTX + corticosteroids or SSZ + HCQ+ intra-articular corticosteroids
7	Initial use of TCZ + MTX or TCZ in monotherapy is significantly superior (in the short and long term) to MTX regarding RA activity (DAS28, DAS28 remission, CDAI, CDAI remission, EULAR and ACR responses, etc.), HAQ, radiographic progression, and quality of life

Abbreviations: IFX = infliximab; ADA = adalimumab; ETN = etanercept; CZP = certolizumab pegol; TCZ = tocilizumab; SSZ = sulphasalazine; HCQ = hydroxychloroquine; MTX = methotrexate; csDMARDs = classical synthetic disease-modifying anti-rheumatic drugs; RA = rheumatoid arthritis; SF-36 = 36-item short-form health survey; ACR = American College of Rheumatology; DAS28 = disease activity score of 28 joints; CEDAI = clinical disease activity index; SDAI = simplified disease activity index; HAQ = health assessment questionnaire; EULAR = European Alliance of Associations for Rheumatology; RAQoL = rheumatoid arthritis quality of life scale.

**Table 2 jcm-13-00048-t002:** Basal characteristics of RA patients in the included studies [18,19,20,21,22,23,24,25,26,27,28,29,30,31,32,33,34].

	Age (Years)	Women	RA Duration	RF+	ACPA+	RA Activity	Structural Damage
IFX	45–55	60–71%	5–7.4 months	61–67%	Hasta 90%	DAS28 > 5	70%
ADA	46–56	63–79%	3–4.4 months	69–74%	-	DAS28 5.5–6.3	SHS 6.3–7.5
ETN	48	76%	7 months	53%	77%	DAS28-CRP 4.17	mTTS 6.69–8.01
CZP	50–52	69–76%	3–7 months	72–97%	-	DAS28 6.7	77.3%
TCZ	53–55	61–76%	1 month	75%	-	DAS28 5.2	SHS 0

Abbreviations: RA = rheumatoid arthritis; RF = rheumatoid factor; ACPA = anti-citrullinated peptide antibodies; IFX = infliximab; ADA = adalimumab; ETN = etanercept; CZP = certolizumab pegol; TCZ = tocilizumab; DAS28 = disease activity score of 28 joints; CRP = C reactive protein; SHS = Sharp-van der Heijde index; mTTS = modified Sharp-van der Heijde index.

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
