# Peer review of "Initial Treatment with Biological Therapy in Rheumatoid Arthritis"

_jcm, 2023, doi:10.3390/jcm13010048_

Round 1

Reviewer 1 Report

Comments and Suggestions for Authors

Comments on the Quality of English Language

Minor editing of English language required

Author Response

Dear reviewer,

First of all, we would like to thank you for your interest and dedication in reviewing the document, and for your constructive comments and suggestions. We really appreciate it. We are confident that with this new version our work has clearly improved. Here we answer to the questions and suggestions.

Best regards,

Dr. Jesús Tornero on behalf of all authors

Initial Treatment with Biological Therapy in Rheumatoid Arthritis

Suggestion: major

  1. Please remove the numbering and labels of ‘(1) Background:, (2)…(4) Conclusions:’ and keep the text.

Thank you so much for the suggestion, we have removed the numbering.

  1. Please add a simple flowchart to indicate your flow of thought for the methods section – that is, the key words search, article selection criteria, final number of articles selected etc.

We have added a simple flowchart in the methods section, right after the selection criteria.  Thanks.

  1. Please update Table 1 by listing the individual studies in each row (nearly 16 rows as you have cited studies 18-34), outlining the total ‘n’ number of patients included in that study, duration of the study, other important technical information of the study, and the main conclusions of that study. As I understand, some of the main conclusions may overlap but that is fine as long as we know what conclusions you have made from which study.

All of this information is in the evidence and results tables in the supplementary material. The tables are quite big, therefore they are in the supplementary material.

  1. How about patients with persistent inflammatory refractive RA and those with refractory but without the inflammation?

Thank you very much for this interesting and relevant question. In this article we have just focused on the indication of initial treatment with biologics. However, we are currently precisely working on the suggested populations and another article will be published in the next months.

  1. Line 340-341, what do you think are the reasons or not considering bDMARD as primary RA treatment course? Toxicity? Contraindications? Pulmonary risks and infections? Please read about this, reflect on its’ negative impact. Then be mindful of the knowledge and then comment on bDMARDs being used as primary RA treatment. Is the result worth the risk? Is there a reason that it is always Safety >>> efficacy? Please reconsider these lines in your manuscript very carefully.

You are right. We have commented on this in the discussion section as follows: “It is striking that national and international consensus documents do not consider the use of bDMARD as an initial treatment for RA, citing various reasons, the foremost of which is cost [10]. Bearing in mind the current context, in which the arrival of biosimilars has brought a considerable reduction in the net cost of bDMARD[13], this position should be reconsidered. But taking into account the risks with the use of bDMARDs, we consider very important to individualize the use of bDMARD as an initial treatment trying to select patients that might benefit most. In our opinion this treatment strategy might be particularly effective in RA patients with poor prognostic factors.”   

  1. Please add a section of conclusions and future directions – reflecting on your work, the positives, negatives and overall approach. Also include what you think the future of Ra holds with respect to the information you have shared in your manuscript.

Thank you so much for your suggestion. We have added a section of conclusions and future directions at the end of the manuscript. This is the new paragraph.

“Conclusions and future directions

Initial treatment with bDMARD in RA patients is effective, with an acceptable safety profile, and in our current context should be considered, particularly for patients with poor prognostic factors. The final decision on RA treatment should be made on a case-by-case basis and in consensus with patients.

On the other hand, more research is needed in order to expedite and optimize treatment stratification by using advanced integrative modelling of complex health data (genetic, biomarkers, clinical data, etc.).”

Reviewer 2 Report

Comments and Suggestions for Authors

1. Kindly follow the PRISMA guidelines and provide a PRISMA checklist.

2. Please provide the complete details of search strategy as an electronic supplementary material.

3. Provide a summary of key characteristics of the included studies in a Table.

4. Provide the strength (Very low, low, moderate, high) for each of the recommendations.

Comments on the Quality of English Language

Minor edits are required.

Author Response

Dear reviewer,

First of all, we would like to thank you for your interest and dedication in reviewing the document, and for your constructive comments and suggestions. We really appreciate it. We are confident that with this new version our work has clearly improved. Here we answer to the questions and suggestions.

Best regards,

Dr. Jesús Tornero on behalf of all authors

  1. Kindly follow the PRISMA guidelines and provide a PRISMA checklist.

Thank you so much for the suggestion. We have followed the PRISMA guidelines and provided the PRISMA checklist along with the other documents.

  1. Please provide a complete details of search strategy as an electronic supplementary material

You are right. We have added the studies flow chart (in the main document) and the details of the search strategy in the supplementary material.

  1. Please provide a summary of key characteristics of the included studies in a table.

This information is available in the supplementary material, thanks.

  1. Please provide the strength (Very low, low, moderate, high) for each of the recommendations.

We agree with you, we have added the strength of the recommendation.

Reviewer 3 Report

Comments and Suggestions for Authors

Dear Author(s),

Thank you for your interesting and clinically relevant manuscript.

The objective, which was to assess the efficacy, effectiveness and safety of initial treatment of RA with bDMARD, is clinically important and interesting.

Introduction is very well-written. I do not have any suggestions for that paragraph. Everything is nicely presented, mentioned, and covered. The aim is clear.

Regarding methodology section, which is nicely performed, there are however few suggestions. Mention if you followed PRISMA guidelines for systematic reviews. Also state why have not you performed meta-analysis (due to heterogeneity or?). Search strategy should be more clearly presented. Add flow diagram.

When presenting results you should distinguish between primary and secondary objectives from studies you are mentioning. Secondary results are only exploratory due to lack of statistical power and thus can lead to invalid conclusions!

When talking about cost-effectiveness report QALYs and ICERs!

Provide comments on external validity from the studies. Also state more limitations that arise from your research within the discussion section.

Conclusions are nicely summarized and based on the results obtained, thus no modification is needed here.

Best regards, Peer-reviewer

Author Response

Dear reviewer,

First of all, we would like to thank you for your interest and dedication in reviewing the document, and for your constructive comments and suggestions. We really appreciate it. We are confident that with this new version our work has clearly improved. Here we answer to the questions and suggestions.

Best regards,

Dr. Jesús Tornero on behalf of all authors

  1. Regarding the review, we actually performed a scoping review. Search strategies are depicted in the supplementary material. A flow chart diagram is also provided in the methods section. We have provided the PRISMA checklist along with the rest of documents. Finally, as you have suggested we did not perform a meta-analysis due to heterogeneity.
  2. Regarding the primary and secondary outcomes, we have highlighted the included studies primary outcome/s in the table 1 of the article. Thank you for your suggestion.
  3. You are right connected to the cost-effectiveness studies. We have clarified this point in the article
  4. We have also commented on the external validity of the studies in the limitations section. Thank you for the suggestion

Round 2

Reviewer 1 Report

Comments and Suggestions for Authors

The authors have addressed all of my queries. I would encourage them to add main conclusions to each of the study in their supplementary table 4.

Comments on the Quality of English Language

Minor editing of English language required

Author Response

Dear reviewer,

We would like to thank you for your interest and time for reviewing a second time our work.

Here we comment on your suggestions.

Best regards,

Dr. Jesús Tornero on behalf of all authors

  1. The authors have addressed all of my queries. I would encourage them to add main conclusions to each of the study in their supplementary table 4.

Thank you for the suggestion. We have added a new column in the table 4 of the supplementary material with the studies main conclusions.

Reviewer 2 Report

Comments and Suggestions for Authors

Nil

Author Response

Dear reviewer,

We would like to thank you for your interest and time for reviewing a second time our work.

Best regards,

Dr. Jesús Tornero on behalf of all authors